# SAR Study and Molecular Mechanism Investigation of Novel Naphthoquinone-furan-2-cyanoacryloyl Hybrids with Antitumor Activity

**DOI:** 10.3390/pharmaceutics14102104

**Published:** 2022-10-01

**Authors:** Pingxian Liu, Dongmei Fan, Wenliang Qiao, Xinlian He, Lidan Zhang, Yunhan Jiang, Tao Yang

**Affiliations:** 1Laboratory of Human Diseases and Immunotherapies, West China Hospital, Sichuan University, Chengdu 610041, China; 2Institute of Immunology and Inflammation, Frontiers Science Center for Disease-Related Molecular Network, West China Hospital, Sichuan University, Chengdu 610041, China; 3Laboratory of Lung Cancer, Lung Cancer Center, West China Hospital, Sichuan University, Chengdu 610041, China; 4State Key Laboratory of Biotherapy and Cancer Center, West China Hospital, Sichuan University, Chengdu 610041, China; 5Department of Cardiovascular Surgery, West China Hospital, Sichuan University, Chengdu 610041, China

**Keywords:** antitumor agent, Naphthoquinone-furan-2-cyanoacryloyl, pharmacophore hybridization, drug discovery

## Abstract

A series of novel naphthoquinone-furan-2-cyanoacryloyl hybrids were designed; they were synthesized and preliminarily evaluated for their anti-proliferative activities in vitro against several cancer cell lines and normal cells. The most potent compound, **5c**, inhibited the proliferation of HeLa cells (IC_50_ value of 3.10 ± 0.02 μM) and colony survival, and it induced apoptosis while having relatively weaker effects on normal cells. Compound **5c** also triggered ROS generation and accumulation, thus partially contributing to the observed cell apoptosis. A Western blotting analysis demonstrated that compound **5c** inhibited the phosphorylation of STAT3. Furthermore, a biolayer interferometry (BLI) analysis confirmed that compound **5c** had a direct effect on STAT3, with a KD value of 13.0 μM. Molecular docking showed that **5c** specifically occupied the subpockets in the SH2 domain, thereby blocking the whole transmission signaling process. Overall, this study provides an important structural reference for the development of effective antitumor agents.

## 1. Introduction

Cancer, a severe human health issue, is one of leading causes of death on a global scale. Despite significant advancements in the treatment of cancer over recent decades, the success of therapy has always been a great challenge [1]. Thus, there is an urgent demand to find effective anticancer drugs to address this human health issue.

In the attempt to find anticancer agents, quinone compounds, an important group of secondary metabolites in plants with anticancer activities, have attracted increasing attention from pharmaceutical enterprises and pharmacists [2]. Because it is an important pharmacophore, quinone scaffolds have been widely applied to develop anticancer agents, including natural and synthesized quinones [3] (Figure 1). Several natural quinones such as doxorubicin, daunorubicin, mitoxantrone, and mitomycin C have been extensively used in treating various cancers [2]. Plumbagin and shikonin, both natural naphthoquinones, have shown potential anticancer activity against breast cancer by inhibiting STAT3 phosphorylation [4,5]. Most notably, the natural product napabucasin was approved by the FDA as an orphan drug for the treatment of gastric/pancreatic cancer in 2016 [6,7]. Aside from the natural quinones, various quinone derivatives have been synthesized, which have good anti-proliferative activity toward tumor cells; atovaquone for example can suppress the viability of various cancer cells [8]. Ly-5 is a potent anticancer compound with IC_50_ values ranging from 0.5 to 1.4 μM against various cancer cell lines, and it has suppressed tumor growth in an in vivo mouse model [9]. Shikonin derivatives obtained by structural modification can dose-dependently induce cell apoptosis in MDA-MB-231 cell lines [10,11]. Generally, the anticancer activities of quinones are mainly associated with several mechanisms, including the inhibition of topoisomerase I and II (Top 1 and Top 2), NAD[P]H-quinone oxidoreductase (NQO1), STAT3 and NF-κB signaling pathways, regulation of the tumor suppressor factor p53, and induction of apoptosis via ERS and DNA damage through the generation of ROS [12]. Therefore, it is promising to discover anticancer agents that bear quinone scaffolds.

Cinnamic acid and its derivatives are important scaffolds of natural products and exert many biological activities, such as anti-proliferative, antineoplastic, and antimetastatic properties [13]. AG490, an analog of cinnamic acid benzyl amide, was shown to inhibit tumor cell growth and increase sensitivity to apoptotic stimuli by inhibiting the JAK2/STAT3 pathway [14]. However, further development of AG490 was suspended due to its instability and lack of potency in biological matrices [15]. Later, a large number of novel cinnamic acid benzyl amide analogs, such as WP1609 and WP1066 (Figure 2), were reported to inhibit the proliferation of cancer cells in vitro and in vivo [16]. WP1609 exhibited anticancer activity by inhibiting the atypical protein kinase Rio1 [17]. WP1066, a well-known cinnamic acid analog, can degrade JAK2 proteins and prevent the downstream signaling and activation of STAT3 and PI3K pathways [15]. In addition, WP1066 has oral biological activity, can pass through the brain barrier, and is currently in clinical phase 3 research [16,18]. As a result, further studies of WP1066 are likely to yield more encouraging outcomes for discovering anticancer agents. In addition, furan derivatives are also important bioactive compounds. For example, dantrolene [19] and nifurtimox [20] were used to treat malignant hyperthermia and Chagas disease, respectively. Recently, nifurtimox has also been found to significantly inhibit the growth of neuroblastoma cell lines in vivo [21].

The privileged fragment combination (PFC) strategy is a common approach used to discover novel drugs [22,23,24]. Herein, as an ongoing effort toward developing effective anticancer drugs, we combined furan and two promising pharmacophores, naphthoquinone [12] and 2-cyanoacryloyl [15,25], to construct naphthoquinone-furan-2-cyanoacryloyl hybrids as a novel scaffold (Figure 3). Based on this scaffold, we synthesized a series of compounds and evaluated their anticancer properties against several cancer cell lines. Then, we conducted cell cycle analysis and apoptosis studies on the most potent compound **5c** in vitro. Additionally, Western blotting and BLI analyses revealed the mechanism of action of this compound. These results indicated that the new hybrid **5c** could be further investigated as an ideal lead compound for antitumor agents.

## 2. Results and Discussion

### 2.1. Chemistry

In this study, thirty novel naphthoquinone-furan-2-cyanoacryloylamide and naphthoquinone-furan carboxamide derivatives were synthesized to evaluate their anticancer effects. The synthesis of target compounds is illustrated in Figure 1. We conducted a treatment of 2-bromo-1,4-dimethoxynaphthalene **1** with 2-formylfuran-5-boronic acid afforded 5-(1,4-dimethoxynaphthalen-2-yl)furan-2-carbaldehyde **2**, and it was reacted with cyanide derivatives (**3a**–**3i**) to yield the intermediates **4a**–**4i**. Then, the (1,4-dimethoxynaphthalen-2-yl)furan intermediates **4a**–**4g** were oxidized to 2-(furan-2-yl)naphthalene-1,4-ditypes (**5a**–**5g**) using diammonium cerium(IV) nitrate as the oxidant. The treatment of 2-bromo-1,4-dimethoxynaphthalene **1** with (5-(methoxycarbonyl)furan-2-yl)boronic acid resulted in the formation of methyl 5-(1,4-dimethoxynaphthalen-2-yl)furan-2-carboxylate **6**, where it was conveniently oxidized and hydrolyzed to provide compound **6a** and the corresponding acid **7,** respectively. To examine the importance of the 2-cyanoacryloyl moiety on the anti-proliferative activity, amide compounds **8a**–**i** and **9a**–**c** were also synthesized. Using HATU as a coupling reagent, acid **7** was reacted with either the aromatic amines or aliphatic amines to afford intermediates **8a**–**8i**. Finally, **8a**–**8c** were oxidized to 2-(furan-2-yl)naphthalene-1,4-diones (**9a**–**9c**) using diammonium cerium(IV) nitrate as the oxidant. 

### 2.2. Biological Evaluation

#### 2.2.1. Evaluation of In Vitro Antitumor Activity and Safety 

First, the inhibition rates at 20 μM for all newly synthesized compounds were evaluated on four cancer cell lines (HeLa, PC3, A549, and HCT116) by a CCK-8 assay. As shown in Table 1, most compounds synthesized in this paper exerted weak to moderate inhibition rates (below 50% inhibition rate) on the test cancer cell lines. A variety of aliphatic amine substituents were well tolerated (**4e**, **4f**, **5c**, and **8f**). These compounds exhibited higher inhibition compared with the aromatic amine substituent derivative **4h**, with relatively low potency (0.25–32.35%). Furthermore, the results showed that the naphthoquinone derivative **5c** exhibited a higher inhibition rate (24.69–75.78%) than naphthalene derivative **4c** (8.99–24.23%). The conversion of 2-cyanoacryloylamide (**4a** and **5a**) to carboxamide (**8a** and **9a**) led to a significant decrease in inhibitory activities, suggesting that 2-cyanoacryloylamide was suitable for inhibitory activity potency.

Among the compounds, **4e** and **8i** exhibited moderate anti-proliferation activity against one or two types of tumor cell lines with inhibition rates above 50%. Specifically, **8c** exhibited a 55.62% inhibition rate against HeLa cells. The inhibition rates of **4f** against HeLa and A549 cells were 59.1% and 59.49%, respectively. The most active compound was compound **5c**, which exhibited 67.67% and 75.78% inhibition rates against the HCT116 and HeLa cells at the test concentrations, respectively.

To further evaluate the anticancer effect of compound **5c**, we evaluated its IC_50_ values against eight selected types of cancer cell lines, with WP1066 as a positive control. To demonstrate whether **5c** would show the expected selectivity between normal cells vs. cancer cells, we also determined the IC_50_ values of **5c** against normal human L02 cells. As presented in Table 2, the IC_50_ values of **5c** were greater than 20.5 μM against six types of cancer cell lines, including PC3, A549, MDA-MB-231, B16, Hep3B and SKOV3. However, it had the best cytotoxicity to HeLa cells, even higher than WP1066. Compound **5c** was also less toxic to normal L02 cells than WP1066 was, suggesting that compared with WP1066, **5c** has stronger cytotoxicity and inhibition of cell proliferation on cervical cancer HeLa cells, as well as a higher safety on normal cells. In addition, the IC_50_ value of **5c** against HeLa cells was 3.10 μM, while it exhibited poor activity against the other seven types of cancer cell lines, suggesting that **5c** exhibited certain selectivity among tumor cells. Therefore, the newly synthesized compound **5c** deserves further investigation.

#### 2.2.2. Compound **5c** Inhibited HeLa Cell Migration

Migration is a key step in cancer progression. Therefore, the antimigration activity of compound **5c** was tested. Cells seeded at a proper density were scratched in the middle of each group. Subsequently, the cells were cultured in a medium with compound **5c** or 0.1% dimethyl sulfoxide (DMSO) for 24 h. As shown in Figure 4A,B, after treatment of HeLa cells with 6 μM of compound **5c** for 48 h, cell migration was significantly inhibited.

#### 2.2.3. Compound **5c** Inhibited the Colony Formation of HeLa Cells

A cell colony formation experiment is an effective method to assess the anti-proliferation ability of compounds. Therefore, a colony formation assay was performed to test whether compound **5c** treatment could inhibit proliferation. As shown in Figure 5A,B, the colony forming ability of HeLa cells was significantly inhibited after compound **5c** treatment, and the result was consistent with the CCK-8 assays.

#### 2.2.4. Compound **5c** Blocks HeLa Cells in G2/M Phase

To investigate the effect of compound **5c** on the cell cycle distribution, a flow cytometric analysis was performed. HeLa cells were treated with **5c** at concentrations of 0 μM, 6 μM, 9 μM, and 12 μM for 24 h and then subjected to a flow cytometric analysis after DNA staining. As shown in Figure 6A, the untreated cells exhibited the expected pattern for continuously growing cells, whereas the cells treated with **5c** progressively increased during the G2/M phase of the cell cycle at 6 μM, 9 μM, and 12 μM. For example, the HeLa cell population gradually increased from 10.19% at 0 μM to 23.88% at 12 μM in the G2/M phase (Figure 6B). Simultaneously, the percentage of G0/G1 and S phase cells significantly decreased. Accordingly, treatment of HeLa cells with **5c** induced a G2/M phase arrest of cell cycle progression.

#### 2.2.5. Compound **5c** Induced the Generation of Intracellular Reactive Oxygen Species

Several quinone derivatives have been reported to up-regulate the intracellular ROS level, which plays a vital role in cell survival. Therefore, we measured the ROS levels in **5c**-exposed HeLa cells. The level of ROS in the HeLa cells treated with **5c** was observed to be higher than those without the treatment of **5c**. The effect of **5c** on HeLa cells under standard culture conditions significantly increased the level of ROS in cancer cells (Figure 7B).

Some studies have shown that oxidative stress can inhibit AKT [26,27,28]. In this study, **5c** effectively inhibited the phosphorylation level of AKT [29,30] (Figure 7C), and the difference in phosphorylation levels was statistically significant (Figure 7D); this subsequently inhibited the growth and proliferation of HeLa cells. This finding is consistent with the result that compound **5c** induced intracellular ROS production.

We then investigated whether **5c** caused proliferation inhibition by causing the accumulation of ROS along with decreased activity of AKT in human HeLa cells. The antioxidant N-acetyl cysteine (NAC) was worked as a ROS scavenger and was applied to further analyze the elevation of ROS. We previously performed a DCFH-DA flow cytometry assay, which showed that exposure of **5c** in HeLa cells augmented the ROS levels (Figure 7A), and which could be markedly suppressed by NAC (5 mM, 1 h) (Figure 7F). Furthermore, we pre-treated HeLa cells with NAC for 1 h. After that, we treated cells with compound **5c** for an additional 48 h. The CCK-8 analysis exhibited that NAC could weaken the inhibitory effect of cell viability induced due to **5c** (Figure 7E). In addition, we also found that pretreatment with NAC reversed the levels of AKT(T308) phosphorylation (Figure 7H). Taken together, the accumulation of ROS had a role in the **5c**-induced proliferation and AKT(T308) phosphorylation inhibition.

#### 2.2.6. Effect of Compound **5c** on HeLa Cell Apoptosis

Apoptosis in cancer cells could be caused by the generation of intracellular ROS [31,32,33]. Three methods were utilized to assess the effect of compound **5c** on the apoptosis of HeLa cells. First, Hoechst 33342 was utilized to stain the HeLa cells that were pretreated with different concentrations of compound **5c**. Then, the cells were observed under a fluorescence microscope. As shown in (Figure 8A), chromatin condensation and chromatinorrhexis were observed in the treated cells. Second, an annexin V-FITC/PI double staining method was used to quantitatively detect the effect of compound **5c** on cell apoptosis. A quantitative analysis was performed using a flow cytometry analysis. The results shown in (Figure 8B) indicate that compound **5c** induced HeLa cell apoptosis in a dose-dependent manner, and the percentage of apoptotic cells reached 46.70% at a dose of 12 μM (Figure 8C). Third, the levels of Bcl-xl family proteins, well-known as apoptosis-related proteins, were determined by Western blotting analysis. The levels of Bax in the HeLa cells increased after treatment with compound **5c** (Figure 8D), while the total Bcl-xl levels significantly decreased (Figure 8E).

#### 2.2.7. Compound **5c** Inhibited STAT3 Y705 Phosphorylation in Cell-Based Assays

As many quinone compounds have anti-proliferation activity by inhibiting the STAT3 [34], we evaluated whether compound **5c** could inhibit its phosphorylation. HeLa cells were incubated with different concentrations (6–12 μM) of **5c** for 24 h. Then, the levels of total STAT3 and phosphorylated STAT3 at the tyrosine 705 residue were detected by a Western blot analysis. As shown in Figure 9, treatment with graded concentrations of compound **5c** dose-dependently abrogated Y705 STAT3 phosphorylation in the HeLa cells, and the level of total STAT3 was slightly down-regulated (Figure 9A), which indicates that compound **5c** may inhibit the phosphorylation of the tyrosine site of STAT3 and inhibit the expression of total STAT3. Furthermore, to explore the relationship between the ROS increase induced by **5c** and the inhibition of STAT3 (Y705) phosphorylation, we pretreated HeLa cells with NAC (5mM, 1 h), and then incubated the HeLa cells with **5c** (12 μM, 24 h). Western blotting results showed that the phosphorylation level of STAT3 (Y705) was reversed by the addition of NAC (Figure 9B), indicating that the compound **5c**-induced STAT3 phosphorylation inhibition may be partially ROS-dependent. This result is consistent with several reported studies that STAT3 inhibition in cancer cells could be partially associated with the increase in ROS level [35,36,37,38].

#### 2.2.8. Kinetic Affinity of Compound **5c** against STAT3

To confirm that compound **5c** is a direct STAT3 inhibitor, a BLI (Biolayer interferometry) analysis was performed. Briefly, recombinant STAT3 with a His-tag was immobilized on SSA biosensors, and the dissociation constants were determined by measuring the binding to serial dilutions of **5c** at concentrations ranging from 7.81 to 250 μM (Figure 10A). The signal was collected, and the kinetic affinity was calculated by the software. The steady-state curve reaches saturation (Figure 10B). The binding affinity (KD) was calculated as 13.0 μM. The results indicate that compound **5c** has a strong binding affinity with the STAT3 protein.

### 2.3. Docking Study of Compound **5c** with the STAT3 SH2 Domain

A large percentage of STAT3 inhibitors reduce STAT3 phosphorylation by occupying the SH2 domain [34,39]. Given that compound **5c** remarkably reduced STAT3 phosphorylation, a molecular docking assay was subsequently performed to investigate its binding model with the STAT3 SH2 domain (PDB: 6NUQ). The investigation showed that the molecule occupied the solvent-accessible subpocket, pY. The carbonyl on the naphthoquinone ring in compound **5c** formed extensive hydrogen bonding interactions with the side chains of Arg609 and backbone amide nitrogen atoms of Glu612. The carbonyl moiety of 2-cyanoacryloyl formed one hydrogen bond with Lys 591 (Figure 11), whereas the morpholine moiety had no specific interactions with any residue in STAT3 and was fully exposed to the solvent environment. Moreover, a suitable conformation of compound **5c** contributed to the tight binding by a considerable amount. The predicted binding energy was −39.68 kcal/mol. This result implied that compound **5c** reduced phosphorylation by binding to the STAT3 SH2 domain.

### 2.4. Molecular Dynamics Simulation Analyses

In order to further study the binding position of potent compound **5c** and STAT3 and verify the accuracy of the docking results, we performed 100 ns MD molecular dynamic simulations using the proposed binding modes of compound **5c** to gain information about their dynamic behavior within the active site of STAT3. The binding free energy ($\Delta G_{\mathrm{bind}}^{\mathrm{cal}}$) for **5c** was −33.87 kcal/mol (Appendix A). RMSDs was also calculated to assess the conformational stability of STAT3 throughout the simulation time. The RMSDs of the two systems remained stable in the last 20 ns for **5c** and last 30 ns for **SI-109** trajectories. The equilibrium time for further analyses was considered after 15 and 20 ns for the **5c** and **SI-109** complexes, respectively (Appendix A). For compound **5c**, the first representative frame of the first cluster with the highest population showed four hydrogen bonds between the naphthoquinone moiety and Arg609, between the furan ring and the two residues Glu612 and Arg609, and between the 2-cyanoacryloyl group and Lys591 (Appendix A).

## 3. Conclusions

In this study, a series of quinone-furan-2-cyanoacryloyl derivatives were synthesized and evaluated for their anti-proliferative effects in vitro. Among the tested compounds, compound **5c** had the best anticancer activity, with IC_50_ values of 3.10 ± 0.02 μM against HeLa cells. Compound **5c** also displayed a higher selectivity than the compound WP1066. The Hoechst 33342 and annexin V-FITC/PI staining experiments proved that **5c** induced the apoptosis of HeLa cells. The protein expression of the pro- and anti-apoptotic proteins Bcl-xl and Bax was also influenced by compound **5c** in a concentration-dependent manner. Compound **5c** increased ROS generation and accumulation in HeLa cells, which could induce cell apoptosis. The Western blotting analysis suggested that compound **5c** reduced the level of activated STAT3. The affinity assay confirmed that compound **5c** is a direct binder of STAT3, with a KD value of 13.0 μM. Docking studies implied the possible binding mode of compound **5c** with the STAT3 SH2 domain. In summary, the newly synthesized quinone-furan-2-cyanoacryloyl hybrids presented in this study provide a structural reference for the development of anticancer agents against the STAT signaling pathway.

## 4. Experimental

### 4.1. Chemistry

All reagents and chemicals are commercially available and were used without further purification. ^1^H and ^13^C NMR spectra were recorded on a Bruker Avance 300 spectrometer (400 MHz and 100 MHz for ^1^H and ^13^C NMR, respectively) in CDCl_3_ or DMSO-*d*_6_. The MS spectra were recorded using an Agilent spectrometer (9575c inert MSD; Agilent Technologies, Santa Clara, CA, USA). The IR spectra were recorded using a Thermo Fisher Fourier transform infrared spectrometer.

5-(1,4-Dimethoxynaphthalen-2-yl)furan-2-carbaldehyde (**2**).

2-formyl-4-methoxyphenyl boronic acid, Cs_2_CO_3_, and Pd(dppf)Cl_2_ were added to a solution of **1** (13.5 g, 50 mmol) in dioxane (100mL) under N_2_ protection. The resulting mixture was stirred at 90 °C overnight. Then, the solution was cooled to room temperature, and the solvents were removed under reduced pressure. The residue was extracted with EtOAc and evaporated in a vacuum. The residue was purified by chromatography on a silica gel chromatography to provide **2**. IR (KBr) 3436.43, 1669.91, 1597.25, 1375.22, 1105.10, 1032.97, 768.09 cm^−1^. ^1^H NMR (400 MHz, DMSO-*d_6_*) δ 9.69 (s, 1H), 8.19 (d, *J* = 8.2 Hz, 1H), 8.13 (d, *J* = 8.2 Hz, 1H), 7.76 (d, J = 3.8 Hz, 1H), 7.68 (dd, *J* = 11.1, 4.0 Hz, 1H), 7.66–7.59 (m, 1H), 7.38 (d, *J* = 3.7 Hz, 1H), 7.27 (s, 1H), 4.06 (s, 3H), 3.86 (s, 3H). ^13^C NMR (100 MHz, DMSO-*d_6_*) δ 178.33, 155.42, 152.09, 151.49, 148.14, 128.85, 128.13, 127.58, 126.89, 126.10, 122.94, 122.59, 117.60, 113.25, 100.80, 61.75, 56.30. HRMS (Q-TOF): calculated for C_17_H_15_O_4_, 283.0965 [M + H], found 283.1050.

General Procedure for the Preparation of **4a**–**4i**. 

Piperidine was added to a solution of compound **2** (0.4 mmol) and the cyanide derivates **3a**–**i** (0.42 mmol) in EtOH (5 mL). The reaction mixture was stirred at room temperature overnight, then the mixture was filtered. The filtered cake was washed with 10 mL of EtOH three times and dried in a vacuum to create the compounds.

**4a**–**4i**.(E)-N-(1-benzylpiperidin-4-yl)-2-cyano-3-(5-(1,4-dimethoxynaphthalen-2-yl)furan-2-yl)acrylamide (**4a**).

IR (KBr) 3426.09, 2925.61, 1673.35, 1600, 1383.07, 1102.29, 1028.41, 765.78 cm^−1^. ^1^H NMR (400 MHz, DMSO-*d_6_*) δ 8.22–8.15 (m, 2H), 8.12 (d, *J* = 8.4 Hz, 1H), 8.00 (s, 1H), 7.72–7.65 (m, 1H), 7.65–7.59 (m, 2H), 7.52 (d, *J* = 3.8 Hz, 1H), 7.39 (d, *J* = 3.8 Hz, 1H), 7.31 (dd, *J* = 9.2, 6.8 Hz, 3H), 7.28–7.21 (m, 2H), 4.02 (s, 3H), 3.88 (s, 3H), 3.69 (m, 1H), 3.47 (s, 2H), 2.81 (d, *J* = 11.6 Hz, 2H), 2.02 (t, *J* = 11.4 Hz, 2H), 1.75 (d, *J* = 9.8 Hz, 2H), 1.61 (dd, *J* = 20.2, 11.6 Hz, 2H). ^13^C NMR (100 MHz, DMSO-*d*_6_) δ 161.05, 154.71, 152.20, 147.77, 147.38, 139.07, 135.18, 129.22 (2C), 128.77, 128.63, 128.06, 127.44, 127.33 (2C), 126.74, 125.69, 122.91, 122.61, 117.77, 114.17, 101.20, 101.14, 62.54, 61.49, 56.40, 52.60 (2C), 48.17, 31.60 (2C). HRMS (Q-TOF): calculated for C_32_H_32_N_3_O_4_, 522.2387 [M + H], found 522.2390.

(E)-3-(5-(1,4-dimethoxynaphthalen-2-yl)furan-2-yl)-2-(1,2,3,4-tetrahydroisoquinoline-2-carbonyl)acrylonitrile (**4b**).

IR (KBr) 3436.44, 2932.93, 1650.76, 1598.97, 1383.68, 1102.96, 801.12 cm^−1^. ^1^H NMR (400 MHz, DMSO-*d_6_*) δ 8.19 (d, *J* = 8.2 Hz, 1H), 8.12 (d, *J* = 8.2 Hz, 1H), 7.81 (s, 1H), 7.69 (t, *J* = 7.2 Hz, 1H), 7.65–7.58 (m, 2H), 7.45 (d, *J* = 3.8 Hz, 1H), 7.39 (d, *J* = 3.6 Hz, 1H), 7.28–7.18 (m, 4H), 4.77 (s, 2H), 4.02 (s, 3H), 3.88 (s, 3H), 3.85 (s, 2H), 2.96 (s, 2H). ^13^C NMR (100 MHz, DMSO-*d*_6_) δ 163.42, 154.47, 152.21, 147.70, 147.67, 136.25, 134.76, 133.24, 129.03, 128.79, 128.06, 127.40, 127.12, 126.98, 126.72 (2C), 125.20, 122.88, 122.61, 117.78, 117.73, 114.04, 101.21, 100.44, 61.44, 56.45, 56.10, 44.42, 28.56. HRMS (Q-TOF): calculated for C_29_H_25_N_2_O_4_, 465.1809 [M + H], found 465.1807.

(E)-2-cyano-3-(5-(1,4-dimethoxynaphthalen-2-yl)furan-2-yl)-N-(2-morpholinoethyl)acrylamide (**4c**).

IR (KBr) 3446.46, 2960.93, 1676.58, 1598.20, 1379.94, 1101.01, 773.13 cm^−1^. ^1^H NMR (400 MHz, DMSO-*d*_6_) δ 8.20 (dd, *J* = 14.4, 6.8 Hz, 2H), 8.12 (d, *J* = 8.2 Hz, 1H), 8.05 (s, 1H), 7.69 (t, *J* = 7.0 Hz, 1H), 7.63 (d, *J* = 7.2 Hz, 1H), 7.60 (s, 1H), 7.55 (d, *J* = 3.8 Hz, 1H), 7.39 (d, *J* = 3.8 Hz, 1H), 4.03 (s, 3H), 3.88 (s, 3H), 3.58 (t, *J* = 4.0 Hz, 4H), 3.36 (dd, *J* = 12.8, 6.8 Hz, 2H), 2.47 (t, *J* = 6.8 Hz, 2H), 2.42 (t, *J* = 4.0 Hz, 4H). ^13^C NMR (100 MHz, DMSO-*d*_6_) δ 160.71, 154.30, 151.68, 147.30, 146.93, 135.02, 128.25, 127.56, 126.96, 126.24, 125.55, 122.41, 122.10, 117.30, 117.23, 113.71, 100.66, 99.83, 66.20 (2C), 61.02, 56.89, 55.89, 53.16 (2C), 36.89. HRMS (Q-TOF): calculated for C_26_H_28_N_3_O_5_, 462.2023 [M + H], found 462.2028.

(E)-2-cyano-3-(5-(1,4-dimethoxynaphthalen-2-yl)furan-2-yl)-N-(2-(thiophen-2-yl)ethyl)acrylamide (**4d**).

IR (KBr) 3430.96, 2928.72, 1671.91, 1600.32, 1382.69, 1102.18, 705.40 cm^−1^. ^1^H NMR (400 MHz, DMSO-*d*_6_) δ 8.49 (t, *J* = 5.6 Hz, 1H), 8.19 (d, *J* = 8.2 Hz, 1H), 8.12 (d, *J* = 8.2 Hz, 1H), 8.05 (s, 1H), 7.69 (t, *J* = 7.6 Hz, 1H), 7.65–7.59 (m, 2H), 7.55 (d, *J* = 3.6 Hz, 1H), 7.40 (d, *J* = 3.6 Hz, 1H), 7.35 (d, *J* = 5.1 Hz, 1H), 6.97 (dd, *J* = 5.0, 3.6 Hz, 1H), 6.92 (d, *J* = 2.8 Hz, 1H), 4.03 (s, 3H), 3.88 (s, 3H), 3.48 (dd, *J* = 13.2, 6.8 Hz, 2H), 3.06 (t, *J* = 7.2 Hz, 2H). ^13^C NMR (100 MHz, DMSO-*d*_6_) δ 160.87, 154.34, 151.69, 147.32, 146.91, 141.21, 135.07, 128.25, 127.56, 126.96 (2C), 126.25, 125.61, 125.25, 124.16, 122.41, 122.11, 117.23 (2C), 113.73, 100.67, 99.84, 61.03, 55.90, 41.37, 28.95. HRMS (Q-TOF): calculated for C_26_H_23_N_2_O_4_S, 459.1373 [M + H], found 459.1372.

(E)-2-cyano-3-(5-(1,4-dimethoxynaphthalen-2-yl)furan-2-yl)-N-(2,2,6,6-tetramethylpiperidin-4-yl)acrylamide (**4e**).

IR (KBr) 3423.88, 2958.91, 1672.49, 1603.82, 1382.13, 1102.37 cm^−1^. ^1^H NMR (400 MHz, CDCl3) δ 7.67 (d, *J* = 8.2 Hz, 1H), 7.55 (d, *J* = 8.2 Hz, 1H), 7.44 (s, 1H), 7.03 (dt, *J* = 14.0, 8.2 Hz, 3H), 6.83–6.71 (m, 3H), 3.79 (s, 1H), 3.51 (d, *J* = 3.8 Hz, 3H), 3.36 (d, *J* = 4.2 Hz, 3H), 1.31 (d, *J* = 12.0 Hz, 2H), 0.81–0.64 (m, 8H), 0.61 (d, *J* = 11.1 Hz, 6H). ^13^C NMR (100 MHz, DMSO-*d*_6_) δ 160.93, 154.73, 152.26, 147.45, 134.93, 128.76, 128.12, 127.43, 126.73, 125.50 (2C), 122.89 (2C), 122.60, 117.79, 114.20, 101.54, 101.20, 61.48, 56.40, 51.01 (2C), 44.25 (2C), 34.97, 29.04 (4C). HRMS (Q-TOF): calculated for C_29_H_34_N_3_O_4_, 488.2544 [M + H], found 488.2570.

(E)-3-(5-(1,4-dimethoxynaphthalen-2-yl)furan-2-yl)-2-(octahydro-1H-isoindole-2-carbonyl)acrylonitrile (**4f**).

IR (KBr) 3435.41, 2927.36, 2858.77, 1635.14, 1381.21, 1103.69, 758.97 cm^−1^. ^1^H NMR (400 MHz, DMSO-*d*_6_) δ 8.18 (d, *J* = 8.2 Hz, 1H), 8.11 (d, *J* = 8.4 Hz, 1H), 7.91 (s, 1H), 7.68 (t, *J* = 7.6 Hz, 1H), 7.61 (d, *J* = 11.8 Hz, 2H), 7.49 (d, *J* = 3.0 Hz, 1H), 7.38 (d, *J* = 2.8 Hz, 1H), 4.36 (dd, *J* = 5.8, 3.6 Hz, 4H), 4.01 (s, 3H), 3.88 (s, 3H), 2.34–2.17 (m, 2H), 1.66–1.27 (m, 8H). ^13^C NMR (100 MHz, DMSO-*d*_6_) δ 162.02 (s), 154.55 (s), 152.20 (s), 147.71 (s), 147.66 (s), 136.57 (s), 128.77 (s), 128.05 (s), 127.41 (s), 126.71 (s), 125.56 (s), 122.88 (s), 122.60 (s), 117.83 (s), 117.78 (s), 114.07 (s), 101.21 (s), 100.84 (s), 61.45 (s), 56.50 (s), 52.43 (s), 51.33 (s), 37.86 (s), 35.43 (s), 25.68 (s), 25.43 (s), 22.70 (s), 19.03 (s). HRMS (Q-TOF): calculated for C_28_H_29_N_2_O_4_, 457.2122 [M + H], found 457.2182.

(E)-2-cyano-3-(5-(1,4-dimethoxynaphthalen-2-yl)furan-2-yl)-N-(3-(trifluoromethyl)benzyl)acrylamide (**4g**).

IR (KBr) 3420.95, 1685.35, 1609.35, 1383.82, 1125.29, 764.64 cm^−1^. ^1^H NMR (400 MHz, DMSO-*d*_6_) δ 9.03 (t, *J* = 5.6 Hz, 1H), 8.18 (d, *J* = 8.2 Hz, 1H), 8.15–8.06 (m, 2H), 7.74–7.58 (m, 7H), 7.55 (d, *J* = 3.0 Hz, 1H), 7.39 (d, *J* = 2.8 Hz, 1H), 4.53 (d, *J* = 5.5 Hz, 2H), 4.03 (s, 3H), 3.88 (s, 3H). ^13^C NMR (100 MHz, DMSO-*d*_6_) δ 161.74, 154.98, 152.19, 147.87, 147.41, 141.06, 135.88, 132.10, 129.86, 129.65, 129.34, 128.75, 128.00, 127.42, 126.79 (s), 126.25, 124.53, 124.15, 123.40, 122.89, 122.60, 117.71, 114.22, 101.19, 100.02, 61.47, 56.33, 43.36. HRMS (Q-TOF): calculated for C_28_H_22_F_3_N_2_O_4_, 507.1526 [M + H], found 507.1581.

(E)-2-cyano-3-(5-(1,4-dimethoxynaphthalen-2-yl)furan-2-yl)-N-phenylacrylamide (**4h**).

IR (KBr) 3414.95, 1683.80, 1596.95, 1538.44, 762.31 cm^−1^. ^1^H NMR (400 MHz, DMSO-*d*_6_) δ 10.24 (s, 1H), 8.23–8.10 (m, 3H), 7.71 (d, *J* = 8.4 Hz, 3H), 7.64 (t, *J* = 9.2 Hz, 2H), 7.58 (s, 1H), 7.45–7.41 (m, 1H), 7.38 (t, *J* = 7.6 Hz, 2H), 7.14 (t, *J* = 7.2 Hz, 1H), 4.05 (s, 3H), 3.90 (s, 3H). ^13^C NMR (100 MHz, DMSO-*d*_6_) δ 160.88 (s), 155.17 (s), 152.22 (s), 147.92 (s), 147.30 (s), 138.87 (s), 135.77 (s), 129.16 (2C), 128.75 (s), 128.02 (s), 127.44 (s), 126.83 (s), 126.35 (s), 124.69 (s), 122.90 (s), 122.62, 121.15 (2C), 120.43, 117.71, 117.60, 114.33, 101.28, 101.18, 61.49, 56.41. HRMS (Q-TOF): calculated for C_26_H_21_N_2_O, 425.1496 [M + H], found 425.1526.

(E)-2-cyano-3-(5-(1,4-dimethoxynaphthalen-2-yl)furan-2-yl)-N-(2-(pyrrolidin-1-yl)ethyl)acrylamide (**4i**).

IR (KBr) 3436.00, 1600.36, 1383.90, 763.52 cm^−1^. ^1^H NMR (400 MHz, DMSO-*d*_6_) δ 8.66 (s, 1H), 8.18 (d, *J* = 8.2 Hz, 2H), 8.12 (d, *J* = 8.2 Hz, 1H), 7.68 (t, *J* = 7.6 Hz, 1H), 7.65–7.58 (m, 2H), 7.54 (d, *J* = 2.4 Hz, 1H), 7.42–7.37 (m, 1H), 4.03 (s, 3H), 3.88 (s, 3H), 3.58 (d, *J* = 5.4 Hz, 2H), 3.20 (s, 6H), 1.92 (s, 4H). ^13^C NMR (100 MHz, DMSO-*d*_6_) δ 162.16, 154.90, 152.19, 147.84, 147.35, 135.60, 128.75, 128.05, 127.46, 126.77, 126.06, 122.90, 122.59, 117.71, 117.66, 114.24, 101.18, 100.52, 61.51, 56.37, 53.62, 45.75, 23.17, 8.90. HRMS (Q-TOF): calculated for C_26_H_27_N_3_O_4_, 445.2002 [M + H], found 446.2126.

General Procedure for the Preparation of **5a**–**g**. 

A solution of diammonium cerium (IV) nitrate in water was added to a solution of intermediates **4a**–**4g** in MeCN; the resulting mixture was stirred at room temperature for 2 h. Then the mixture was filtered, and the filtered cake was washed with 10 mL of water three times and dried in a vacuum to give the target compound **5a**–**5g**.

(E)-N-(1-benzylpiperidin-4-yl)-2-cyano-3-(5-(1,4-dioxo-1,4-dihydronaphthalen-2-yl)furan-2-yl)acrylamide (**5a**).

IR (KBr) 3435.70, 2926.61, 1655.32, 1384.37, 1259.47, 781.38 cm^−1^. ^1^H NMR (400 MHz, CDCl3) δ 8.05–7.99 (m, 1H), 7.99–7.94 (m, 1H), 7.91 (s, 1H), 7.70–7.63 (m, 3H), 7.40 (s, 1H), 7.20 (d, *J* = 4.4 Hz, 4H), 7.18–7.08 (m, 3H), 3.81 (dd, *J* = 11.1, 7.5 Hz, 1H), 3.48 (s, 2H), 2.83 (d, *J* = 11.5 Hz, 3H), 2.13 (t, *J* = 10.9 Hz, 2H), 1.83 (d, *J* = 10.6 Hz, 2H), 1.59 (dd, *J* = 20.5, 11.0 Hz, 2H). ^13^C NMR (100 MHz, DMSO-*d*_6_) δ 184.54 (s), 182.57 (s), 161.03 (s), 150.69 (s), 150.05 (s), 135.10 (s), 134.96 (s), 134.72 (s), 134.41 (s), 132.41 (s), 132.14 (s), 131.46 (s), 130.94 (s), 129.28 (s), 126.89 (s), 126.09 (s), 124.64 (s), 120.94 (s), 116.84 (s), 104.29 (s), 60.15 (s), 51.50 (s), 51.45 (s), 29.50 (s). HRMS (Q-TOF): calculated for C_30_H_26_N_3_O_4_, 491.1845 [M + H], found 492.1916.

(E)-3-(5-(1,4-dioxo-1,4-dihydronaphthalen-2-yl)furan-2-yl)-2-(1,2,3,4-tetrahydroisoquinoline-2-carbonyl)acrylonitrile (**5b**).

IR (KBr) 3435.48, 1637.19, 1384.34, 1259.42, 724.77 cm^−1^. ^1^H NMR (400 MHz, DMSO-*d*_6_) δ 8.11 (dd, *J* = 5.8, 3.2 Hz, 1H), 8.06–8.01 (m, 1H), 7.95–7.89 (m, 2H), 7.85 (s, 1H), 7.75 (d, *J* = 3.8 Hz, 1H), 7.42 (d, *J* = 3.7 Hz, 1H), 7.40 (s, 1H), 7.24 (s, 1H), 7.22 (s, 3H), 4.75 (s, 2H), 3.83 (s, 2H), 2.95 (s, 2H). ^13^C NMR (100 MHz, CDCl_3_) δ 184.74, 182.98, 162.92, 162.84, 150.82, 150.62, 136.98, 134.69, 134.59, 134.31, 132.46, 131.72, 129.14, 127.36, 127.14, 126.98, 126.83, 126.76, 126.53, 122.58, 120.89, 116.42, 104.81, 54.43, 48.49, 29.99. HRMS (Q-TOF): calculated for C_27_H_19_N_2_O_4_, 435.1339 [M + H], found 435.1337. 

(E)-2-cyano-3-(5-(1,4-dioxo-1,4-dihydronaphthalen-2-yl)furan-2-yl)-N-(2-morpholinoethyl)acrylamide (**5c**).

IR (KBr) 3435.17, 1629.25, 1550.99, 1384.58, 1113.94, 618.50 cm^−1^. ^1^H NMR (400 MHz, DMSO-*d*_6_) δ 8.58 (s, 1H), 8.12 (s, 1H), 8.10 (dd, *J* = 5.8, 3.2 Hz, 1H), 8.04–7.98 (m, 1H), 7.94–7.88 (m, 2H), 7.76 (d, *J* = 3.8 Hz, 1H), 7.56 (d, *J* = 3.8 Hz, 1H), 7.35 (s, 1H), 3.76 (s, 4H), 3.56 (d, *J* = 3.6 Hz, 2H), 3.17 (s, 6H). ^13^C NMR (100 MHz, DMSO-*d*_6_) δ 184.02, 182.02, 160.86, 150.36, 149.60, 135.21, 134.47, 134.24, 133.86, 131.88, 131.61, 130.50, 126.39, 125.59, 124.79, 120.49, 116.34, 102.78, 63.80, 55.38, 51.68, 36.73. HRMS (Q-TOF): calculated for C_24_H_22_N_3_O_5_, 432.1554 [M + H], found 432.1553. 

(E)-2-cyano-3-(5-(1,4-dioxo-1,4-dihydronaphthalen-2-yl)furan-2-yl)-N-(2-(thiophen-2-yl)ethyl)acrylamide (**5d**).

IR (KBr) 3436.28, 2927.20, 1655.50, 1384.38, 1259.77, 725.37 cm^−1^. ^1^H NMR (400 MHz, DMSO-*d*_6_) δ 8.60 (t, *J* = 5.6 Hz, 1H), 8.15–8.06 (m, 2H), 8.06–8.00 (m, 1H), 7.98–7.88 (m, 2H), 7.75 (d, *J* = 3.6 Hz, 1H), 7.53 (d, *J* = 3.8 Hz, 1H), 7.41–7.33 (m, 2H), 6.97 (dd, *J* = 5.0, 3.5 Hz, 1H), 6.92 (d, *J* = 2.9 Hz, 1H), 3.48 (dd, *J* = 13.0, 7.1 Hz, 2H), 3.06 (t, *J* = 7.1 Hz, 2H). ^13^C NMR (100 MHz, CDCl_3_) δ 189.15, 165.29, 155.35, 154.92, 145.74, 140.94, 140.16, 139.19, 139.02, 138.87, 136.80, 136.71, 135.92, 131.76, 131.53, 130.78, 130.10, 128.62, 128.22, 125.51, 121.05, 108.17, 46.74, 34.12. HRMS (Q-TOF): calculated for C_24_H_17_N_2_O_4_S, 429.0904 [M + H], found 429.0901.

(E)-2-cyano-3-(5-(1,4-dioxo-1,4-dihydronaphthalen-2-yl)furan-2-yl)-N-(2,2,6,6-tetramethylpiperidin-4-yl)acrylamide (**5e**).

IR (KBr) 3383.55, 1673.81, 1256.48, 1190.54, 723.17 cm^−1^. ^1^H NMR (400 MHz, CDCl_3_) δ 8.62 (s, 1H), 8.05–7.98 (m, 1H), 7.94 (d, *J* = 2.9 Hz, 1H), 7.88 (s, 1H), 7.74 (d, *J* = 6.5 Hz, 1H), 7.66 (dd, *J* = 5.8, 3.0 Hz, 3H), 7.39 (s, 1H), 7.16 (d, *J* = 3.4 Hz, 1H), 4.30 (s, 1H), 1.89–1.78 (m, 2H), 1.63 (t, *J* = 12.8 Hz, 2H), 1.38 (s, 6H), 1.32 (s, 6H). ^13^C NMR (100 MHz, DMSO-*d*_6_) δ 184.59, 182.59, 161.00, 150.73, 149.98, 135.01, 134.76, 134.42, 132.40, 132.16, 130.96, 126.91, 126.10, 125.47, 124.70, 120.97, 116.75, 104.24, 57.17 (2C), 41.48 (2C), 30.33, 24.71 (4C). HRMS (Q-TOF): calculated for C_27_H_28_N_3_O_4_, 458.2074 [M + H], found 458.2084.

(E)-3-(5-(1,4-dioxo-1,4-dihydronaphthalen-2-yl)furan-2-yl)-2-(octahydro-1H-isoindole-2-carbonyl)acrylonitrile (**5f**).

IR (KBr) 3439.57, 2929.88, 1647.60, 1258.55, 778.15 cm^−1^. ^1^H NMR (400 MHz, CDCl_3_) δ 7.88 (dt, *J* = 7.2, 3.6 Hz, 1H), 7.81 (dd, *J* = 5.7, 3.3 Hz, 1H), 7.59–7.54 (m, 3H), 7.52 (d, *J* = 3.8 Hz, 1H), 7.34 (s, 1H), 7.04 (d, *J* = 3.8 Hz, 1H), 3.56 (dd, *J* = 10.3, 6.8 Hz, 1H), 3.40 (dd, *J* = 10.3, 5.6 Hz, 1H), 2.07 (ddd, *J* = 17.4, 11.5, 6.0 Hz, 2H), 1.47–1.09 (m, 10H). ^13^C NMR (100 MHz, CDCl_3_) δ 189.33, 187.28, 179.09, 166.32, 155.22, 154.98, 141.50, 139.25, 138.83, 136.79, 136.68, 135.89, 131.46, 130.75, 127.75, 125.38, 120.84, 108.70, 57.33, 56.11, 42.75, 40.10, 30.16, 30.03, 27.21, 27.03. HRMS (Q-TOF): calculated for C_26_H_23_N_2_O_4_, 427.1652 [M + H], found 427.1626, C_26_H_22_N_2_NaO_4_, 449.1477 [M + Na], 449.1504.

(E)-2-cyano-3-(5-(1,4-dioxo-1,4-dihydronaphthalen-2-yl)furan-2-yl)-N-(3-(trifluoromethyl)phenyl)acrylamide (**5g**).

IR (KBr) 3434.81, 1672.24, 1331.26, 1115.89, 702.03 cm^−1^. ^1^H NMR (400 MHz, DMSO-*d*_6_) δ 8.69 (s, 1H), 8.01 (s, 2H), 7.77 (d, *J* = 15.2 Hz, 2H), 7.57 (d, *J* = 9.5 Hz, 3H), 7.48 (s, 4H), 7.29 (s, 1H), 3.52 (s, 2H). ^13^C NMR (101 MHz, DMSO) δ 182.48, 180.58, 180.18, 159.24, 148.70, 148.07, 138.70, 133.58, 132.84, 132.60, 132.37, 130.36, 130.16, 130.02, 128.88, 127.79, 123.99, 122.93, 122.56, 122.35, 122.12 118.98, 116.43, 114.77, 101.48, 41.22. HRMS (Q-TOF): calculated for C_26_H_16_F_3_N_2_O_4_, 477.1057 [M + H], found 477.1010.

Methyl 5-(1,4-dimethoxynaphthalen-2-yl)furan-2-carboxylate (**6**). 

Compound **6** was prepared according to the synthetic route of compound **2** except (5-formylfuran-2-yl)boronic acid was replaced with (5-(methoxycarbonyl)furan-2-yl)boronic acid. IR (KBr) 3439.25, 2929.79, 1726.30, 1142.93, 756.14 cm^−1^. ^1^H NMR (400 MHz, DMSO-*d*_6_) δ 8.18 (d, *J* = 8.2 Hz, 1H), 8.11 (d, *J* = 8.2 Hz, 1H), 7.67 (dd, *J* = 11.1, 4.2 Hz, 1H), 7.64–7.57 (m, 1H), 7.52 (d, *J* = 3.6 Hz, 1H), 7.27 (d, *J* = 3.6 Hz, 1H), 7.23 (s, 1H), 4.05 (s, 3H), 3.88 (s, 3H), 3.84 (s, 3H). HRMS (Q-TOF): calculated for C_18_H_17_O_5_, 313.1071 [M + H], found 313.1069.

Methyl 5-(1,4-dioxo-1,4-dihydronaphthalen-2-yl)furan-2-carboxylate (**6a**).

Compound **6a** was prepared according to the synthetic route of compound **5a**−**5g** except **5a**–**5g** was replaced with compound **6**. IR (KBr) 3437.02, 1741.97, 1293.39, 1208.73, 724.28 cm^−1^. ^1^H NMR (400 MHz, DMSO-*d*_6_) δ 8.09 (dd, *J* = 6.0, 3.0 Hz, 1H), 8.05–7.98 (m, 1H), 7.91 (dd, *J* = 5.2, 3.8 Hz, 2H), 7.60 (d, *J* = 3.6 Hz, 1H), 7.50 (d, *J* = 3.6 Hz, 1H), 7.23 (s, 1H), 3.88 (s, 3H). HRMS (Q-TOF): calculated for C_16_H_11_O_5_, 283.0601 [M + H], found 283.0599.

5-(1,4-Dimethoxynaphthalen-2-yl)furan-2-carboxylic acid (**7**). 

LiOH was added to solution **6** in MeOH/H_2_O (5/1). The resulting mixture was stirred at room temperature overnight, followed by an acidification through the adding of dilute hydrochloric acid. The resulting mixture was filtered and washed with water three times, and the filtration was dried in a vacuum to give solution **7,** which was used in the next step without further purification.

General Procedure for the Preparation of **8a-i**. 

Solution **7** (0.20 mmol) and triethylamine (0.31 mmol) in DMF (3 mL) was stirred with HATU (0.31 mmol) at room temperature for 10 min, and then substituted alkyl amines or N-heterocycles (0.36 mmol) were added. Afterward, the resulting mixture was stirred at room temperature for 2 h, and the mixture was extracted with AcOEt and evaporated in a vacuum, then purified by chromatography using silica gel chromatography to create the corresponding amide.

N-(1-benzylpiperidin-4-yl)-5-(1,4-dimethoxynaphthalen-2-yl)furan-2-carboxamide (**8a**).

IR (KBr) 3434.97, 3284.89, 2937.09, 1629.85, 1376.68, 1105.44, 779.20 cm^−1^. ^1^H NMR (400 MHz, CDCl_3_) δ 8.25 (d, *J* = 8.2 Hz, 1H), 8.11 (d, *J* = 8.4 Hz, 1H), 7.58 (t, *J* = 7.6 Hz, 1H), 7.52 (t, *J* = 7.6 Hz, 1H), 7.38–7.31 (m, 4H), 7.31–7.23 (m, 3H), 7.19–7.11 (m, 2H), 6.31 (d, *J* = 7.6 Hz, 1H), 4.07 (d, *J* = 1.8 Hz, 4H), 3.85 (d, *J* = 2.0 Hz, 3H), 3.59 (s, 2H), 2.94 (d, *J* = 9.6 Hz, 2H), 2.26 (t, *J* = 11.4 Hz, 2H), 2.14–2.00 (m, 2H), 1.86–1.63 (m, 2H). ^13^C NMR (101 MHz, CDCl_3_) δ 157.81, 152.23, 152.21, 147.17, 146.34, 129.34, 129.02, 128.37 (2C), 127.37, 127.24, 126.74, 126.34, 122.50, 122.36, 117.91, 116.82, 112.10, 100.65, 63.02, 60.95, 55.84 (2C), 52.28, 46.08, 32.21, 29.72. HRMS (Q-TOF): calculated for Chemical C_29_H_31_N_2_O_4_, 471.2278 [M + H], found 471.2349.

5-(1,4-Dimethoxynaphthalen-2-yl)-N-(1-phenylethyl)furan-2-carboxamide (**8b**).

IR (KBr) 3435.62, 1626.61, 1375.24, 1105.73, 763.53, 700.58 cm^−1^. ^1^H NMR (400 MHz, CDCl_3_) δ 8.24 (d, *J* = 7.9 Hz, 1H), 8.11 (d, *J* = 8.2 Hz, 1H), 7.58 (dd, *J* = 11.1, 4.2 Hz, 1H), 7.55–7.48 (m, 1H), 7.45 (d, *J* = 7.4 Hz, 2H), 7.38 (t, *J* = 7.6 Hz, 2H), 7.31 (d, *J* = 7.2 Hz, 1H), 7.28 (d, *J* = 3.6 Hz, 1H), 7.15 (d, *J* = 3.6 Hz, 1H), 7.12 (s, 1H), 6.58 (d, *J* = 8.0 Hz, 1H), 5.47–5.28 (m, 1H), 4.04 (s, 3H), 3.84 (s, 3H), 1.67 (d, *J* = 6.8 Hz, 3H). ^13^C NMR (100 MHz, DMSO-*d*_6_) δ 157.56, 151.99, 151.63, 146.94, 146.73, 144.96, 128.89, 128.75 (2C), 127.87, 127.19, 126.89, 126.59 (2C), 126.33, 122.68, 122.54, 118.35, 116.57, 112.29, 101.72, 61.20, 56.42, 48.35, 22.43. HRMS (Q-TOF): calculated for C_25_H_24_NO_4_, 402.1700 [M + H], found 402.1696.

5-(1,4-Dimethoxynaphthalen-2-yl)-N-(3-morpholino-3-oxopropyl)furan-2-carboxamide (**8c**).

IR (KBr) 3435.66, 2930.98, 1666.91, 1101.41, 764.83 cm^−1^. ^1^H NMR (400 MHz, CDCl_3_) δ 8.28–8.20 (m, 1H), 8.10 (d, *J* = 7.9 Hz, 1H), 7.57 (ddd, *J* = 8.2, 6.8, 1.3 Hz, 1H), 7.54–7.47 (m, 2H), 7.25 (dd, *J* = 3.6, 1.8 Hz, 2H), 7.16 (d, *J* = 3.6 Hz, 1H), 4.09 (s, 3H), 3.85 (d, *J* = 8.3 Hz, 3H), 3.79 (dd, *J* = 11.4, 6.0 Hz, 2H), 3.70–3.64 (m, 4H), 3.64–3.58 (m, 2H), 3.52–3.37 (m, 2H), 2.64 (t, *J* = 5.6 Hz, 2H). ^13^C NMR (100 MHz, CDCl_3_) δ 166.47, 154.59, 148.24, 143.08, 142.24, 125.01, 123.20, 122.75, 122.30, 118.54 (2C), 118.39, 114.06, 112.71, 108.04, 96.75, 62.82, 62.54, 56.95, 51.89, 41.74, 37.87, 30.79, 28.84. HRMS (Q-TOF): calculated for C_24_H_27_N_2_O_6_, 439.1864 [M + H], found 439.1891.

5-(1,4-Dimethoxynaphthalen-2-yl)-N-(2-fluoro-5-(trifluoromethyl)benzyl)furan-2-carboxamide (**8d**).

IR (KBr) 3289.76, 1651.21, 1330.15, 1110.27, 773.94 cm^−1^. ^1^H NMR (400 MHz, DMSO-*d_6_*) δ 9.16 (t, *J* = 5.9 Hz, 1H), 8.18 (d, *J* = 8.0 Hz, 1H), 8.10 (d, *J* = 8.2 Hz, 1H), 7.83–7.73 (m, 2H), 7.72–7.63 (m, 1H), 7.63–7.55 (m, 1H), 7.52–7.45 (m, 2H), 7.39 (d, *J* = 3.6 Hz, 1H), 7.25 (d, *J* = 3.6 Hz, 1H), 4.64 (d, *J* = 5.8 Hz, 2H), 4.08 (s, 3H), 3.84 (s, 3H). ^13^C NMR (100 MHz, DMSO-*d*_6_) δ 158.47, 152.04, 151.87, 146.85, 146.50, 128.85, 128.27, 128.11, 127.88, 127.33, 126.95, 126.38, 125.71, 123.00, 122.69, 122.54, 118.23, 117.19, 117.12, 116.89, 112.47, 101.58, 61.19, 56.51, 36.29. HRMS (Q-TOF): calculated for C_25_H_20_F_4_NO_4_, 474.1323 [M + H], found 474.1347.

5-(1,4-Dimethoxynaphthalen-2-yl)-N-(3-(4-methylpiperazin-1-yl)phenyl)furan-2-carboxamide (**8e**).

IR (KBr) 3434.65, 1653.53, 1496.70, 1103.86, 997.29, 769.51 cm^−1^. ^1^H NMR (400 MHz, DMSO-*d*_6_) δ 10.14 (s, 1H), 8.22 (d, *J* = 8.2 Hz, 1H), 8.14 (d, *J* = 8.2 Hz, 1H), 7.70 (t, *J* = 7.2 Hz, 1H), 7.63 (t, *J* = 7.2 Hz, 1H), 7.57 (d, *J* = 4.0 Hz, 2H), 7.48 (s, 1H), 7.33 (d, *J* = 3.6 Hz, 1H), 7.24 (d, *J* = 5.0 Hz, 2H), 6.76 (d, *J* = 3.6 Hz, 1H), 4.13 (s, 3H), 3.89 (s, 3H), 3.23–3.11 (m, 4H), 2.52–2.47 (m, 4H), 2.26 (s, 3H). ^13^C NMR (100 MHz, DMSO-*d_6_*) δ 156.50, 152.18, 152.04, 151.87, 146.92, 146.70, 139.65, 129.52, 128.87, 127.95, 127.05, 126.41, 122.66, 118.25, 117.67, 112.63, 111.82, 108.18, 101.69, 66.83, 61.33, 56.54, 55.07, 48.61, 46.24. HRMS (Q-TOF): calculated for C_28_H_30_N_3_O_4_, 472.2231 [M + H], found 472.2236.

N-(4,4-difluorocyclohexyl)-5-(1,4-dimethoxynaphthalen-2-yl)furan-2-carboxamide (**8f**).

IR (KBr) 3313.35, 1634.62, 1593.53, 1118.65, 770.02 cm^−1^. ^1^H NMR (400 MHz, DMSO-*d*_6_) δ 8.31 (d, *J* = 7.8 Hz, 1H), 8.17 (d, *J* = 8.2 Hz, 1H), 8.09 (d, *J* = 8.2 Hz, 1H), 7.66 (t, *J* = 7.2 Hz, 1H), 7.58 (t, *J* = 7.4 Hz, 1H), 7.45 (s, 1H), 7.33 (d, *J* = 3.6 Hz, 1H), 7.21 (d, *J* = 3.6 Hz, 1H), 4.08 (s, 3H), 4.05–3.97 (m, 1H), 3.83 (s, 3H), 2.16–1.84 (m, 6H), 1.80–1.63 (m, 2H). ^13^C NMR (100 MHz, DMSO-*d_6_*) δ 157.64, 151.99, 151.50, 146.94, 146.73, 128.88, 127.91, 126.95, 126.33, 122.70, 122.54, 118.32, 116.58, 112.32, 101.84, 61.24, 56.55, 45.99, 32.21, 28.52, 28.43. HRMS (Q-TOF): calculated for C_23_H_24_F_2_NO_4_,416.1668 [M + H], found 416.1707, C_23_H_23_F_2_NNaO, 438.1493 [M + Na], found 438.1515.

N-(3-(1H-imidazol-1-yl)propyl)-5-(1,4-dimethoxynaphthalen-2-yl)furan-2-carboxamide (**8g**).

IR (KBr) 3436.75, 2936.87, 1651.78, 1373.56, 1102.08, 760.78 cm^−1^. ^1^H NMR (400 MHz, DMSO-*d*_6_) δ 8.60 (t, *J* = 5.8 Hz, 1H), 8.17 (d, *J* = 8.2 Hz, 1H), 8.08 (d, *J* = 8.4 Hz, 1H), 7.69 (s, 1H), 7.65 (t, *J* = 7.5 Hz, 1H), 7.58 (t, *J* = 7.4 Hz, 1H), 7.47 (s, 1H), 7.30 (d, *J* = 3.5 Hz, 1H), 7.27–7.19 (m, 2H), 6.90 (s, 1H), 4.08 (s, 3H), 4.05 (t, *J* = 6.8 Hz, 2H), 3.82 (s, 3H), 3.29 (dd, *J* = 12.9, 6.4 Hz, 2H), 2.09–1.92 (m, 2H). ^13^C NMR (100 MHz, DMSO-*d*_6_) δ 157.80, 151.53, 150.93, 146.48, 146.21, 137.33, 128.34, 127.42, 126.44, 125.79, 122.19, 122.04, 120.85, 119.36, 117.81, 116.10, 111.93, 101.15, 60.65, 56.14, 43.80, 35.97, 31.00. HRMS (Q-TOF): calculated for C_23_H_24_N_3_O_4_, 406.1761 [M + H], found 406.1757.

5-(1,4-Dimethoxynaphthalen-2-yl)-N-(4-((4-methylpiperazin-1-yl)methyl)-3-(trifluoromethyl)phenyl)furan-2-carboxamide (**8h**).

IR (KBr) 3495.65, 2937.58, 1656.86, 1334.36, 1106.34, 768.71 cm^−1^. ^1^H NMR (400 MHz, DMSO-*d*_6_) δ 10.54 (s, 1H), 8.24 (s, 1H), 8.20 (d, *J* = 8.4 Hz, 1H), 8.12 (d, *J* = 8.3 Hz, 1H), 8.05 (d, *J* = 8.4 Hz, 1H), 7.74 (d, *J* = 8.4 Hz, 1H), 7.67 (t, *J* = 7.4 Hz, 2H), 7.60 (t, *J* = 6.6 Hz, 2H), 7.53 (s, 1H), 7.32 (d, *J* = 3.2 Hz, 1H), 4.11 (s, 3H), 3.87 (s, 3H), 3.59 (s, 2H), 3.46 (s, 8H), 2.44 (s, 8H), 2.25 (s, 3H). ^13^C NMR (100 MHz, DMSO-*d*_6_) δ 162.77, 156.76, 152.57, 152.06, 147.08, 146.15, 140.09, 138.16, 131.82, 128.87, 127.99, 127.13, 126.49, 124.21, 123.43, 122.79, 122.58, 118.36, 118.12, 112.73, 109.60, 101.65, 61.40, 57.80, 56.52 (2C), 54.97 (2C), 52.77, 45.75. HRMS (Q-TOF): calculated for C_30_H_31_F_3_N_3_O_4_, 554.2261 [M + H], found 554.2348.

5-(1,4-Dimethoxynaphthalen-2-yl)-N-(4-(4-(dimethylamino)piperidin-1-yl)-2-methoxyphenyl)furan-2-carboxamide (**8i**).

IR (KBr) 3420.75, 1665.94, 1460.75, 838.38 cm^−1^. ^1^H NMR (400 MHz, DMSO-*d*_6_) δ 9.35 (s, 1H), 8.19 (d, *J* = 8.2 Hz, 1H), 8.12 (d, *J* = 8.4 Hz, 1H), 7.71–7.55 (m, 3H), 7.49 (s, 1H), 7.42 (d, *J* = 3.2 Hz, 1H), 7.29 (d, *J* = 3.6 Hz, 1H), 6.72 (s, 1H), 6.58 (d, *J* = 8.6 Hz, 1H), 4.10 (s, 3H), 3.98–3.81 (m, 8H), 3.34 (d, *J* = 14.2 Hz, 2H), 2.85–2.69 (m, 7H), 2.06 (d, *J* = 11.0 Hz, 2H), 1.70 (dd, *J* = 20.9, 11.6 Hz, 2H). ^13^C NMR (100 MHz, DMSO-*d*_6_) δ 156.25 (s), 152.82 (s), 152.04 (s), 151.91 (s), 149.39 (s), 146.90 (s), 146.82 (s), 128.90 (s), 127.94 (s), 127.02 (s), 126.38 (s), 125.28 (s), 122.74 (s), 122.55 (s), 118.51 (s), 118.16 (s), 117.39 (s), 112.68 (s), 107.76 (s), 101.47 (s), 100.92 (s), 63.10 (s), 61.33 (s), 56.42 (s), 56.24 (s), 48.02 (s), 39.89 (s), 26.05 (s). HRMS (Q-TOF): calculated for Chemical C_31_H_36_N_3_O_5_, 530.2649 [M + H], found 530.2719.

N-(1-benzylpiperidin-4-yl)-5-(1,4-dioxo-1,4-dihydronaphthalen-2-yl)furan-2-carboxamide (**9a**).

Compound **9a** was prepared according to the synthetic route of compound **5a**–**5g** except **5a**–**5g** was replaced with compound **8a**. IR (KBr) 3435.38, 1673.80, 1301.67, 1016.33, 721.53 cm^−1^. ^1^H NMR (400 MHz, MeOD) δ 8.05 (dt, *J* = 9.7, 3.6 Hz, 1H), 8.01–7.96 (m, 1H), 7.74–7.67 (m, 2H), 7.61 (s, 1H), 7.56 (d, *J* = 3.8 Hz, 1H), 7.55–7.49 (m, 1H), 7.25 (d, *J* = 2.8 Hz, 3H), 7.24–7.16 (m, 2H), 7.14 (d, *J* = 3.7 Hz, 1H), 4.02–3.85 (m, 1H), 3.52 (s, 2H), 2.88 (d, *J* = 28.0 Hz, 2H), 2.73–2.71 (m, 2H), 2.15 (t, *J* = 11.0 Hz, 2H), 1.89 (d, *J* = 10.0 Hz, 2H), 1.69 (qd, *J* = 12.2, 3.6 Hz, 2H). ^13^C NMR (100 MHz, DMSO) δ 185.72, 183.37, 158.03, 150.33, 148.37, 138.11, 135.62, 134.96, 134.73, 132.98, 132.70, 130.75, 130.10, 129.09 (2C), 128.13, 127.59, 126.72, 120.76, 117.42, 63.44, 53.17 (2C), 47.33, 39.36, 32.34. HRMS (Q-TOF): calculated for C_27_H_25_N_2_O_4_, 441.1809 [M + H], found 441.1910.

5-(1,4-Dioxo-1,4-dihydronaphthalen-2-yl)-N-(1-phenylethyl)furan-2-carboxamide (**9b**).

Compound **9b** was prepared according to the synthetic route of compound **5a**–**5g** except **5a**–**5g** was replaced with compound **8b**. IR (KBr) 3434.67, 1641.52, 1384.44, 1257.35, 700.53 cm^−1^. ^1^H NMR (400 MHz, DMSO-*d_6_*) δ 9.13 (d, *J* = 8.2 Hz, 1H), 8.09 (dd, *J* = 5.8, 3.2 Hz, 1H), 8.02 (dt, *J* = 7.2, 3.6 Hz, 1H), 7.95–7.87 (m, 3H), 7.59 (d, *J* = 3.6 Hz, 1H), 7.41 (d, *J* = 7.4 Hz, 2H), 7.36 (t, *J* = 7.6 Hz, 2H), 7.27 (d, *J* = 3.6 Hz, 1H), 7.24 (d, *J* = 7.2 Hz, 1H), 5.28–5.17 (m, 1H), 1.55 (d, *J* = 7.2Hz, 3H). ^13^C NMR (100 MHz, DMSO-*d_6_*) δ 185.17, 182.82, 156.75, 149.57, 148.00, 144.65, 134.88, 134.85, 134.64, 132.52, 132.14, 130.49, 128.80 (2C), 127.25, 126.86, 126.55 (2C), 125.99, 119.72, 117.02, 48.18, 22.31. HRMS (Q-TOF): calculated for C_23_H_18_NO_4_, 372.1230 [M + H], found 372.1224

5-(1,4-Dioxo-1,4-dihydronaphthalen-2-yl)-N-(3-morpholino-3-oxopropyl)furan-2-carboxamide (**9c**).

Compound **9c** was prepared according to the synthetic route of compound **5a**–**5g** except **5a**–**5g** was replaced with compound **8c**. IR (KBr) 3436.28, 1655.30, 1261.44, 1113.19, 767.54 cm^−1^. ^1^H NMR (400 MHz, DMSO-*d_6_*) δ 8.93 (t, *J* = 5.6 Hz, 1H), 8.08 (dd, *J* = 5.8, 3.2 Hz, 2H), 8.01 (dd, *J* = 5.8, 3.3 Hz, 1H), 7.93–7.85 (m, 1H), 7.79 (s, 1H), 7.57 (d, *J* = 3.6 Hz, 1H), 7.23 (d, *J* = 3.6 Hz, 1H), 3.54 (dd, *J* = 9.6, 5.2 Hz, 4H), 3.47 (dt, *J* = 9.2, 5.8 Hz, 6H), 2.62 (t, *J* = 7.2 Hz, 2H). ^13^C NMR (100 MHz, DMSO-*d_6_*) δ 182.31, 169.06, 157.03, 149.07, 147.29, 134.37, 134.16, 132.01 (2C), 131.63 (2C), 126.37 (2C), 125.49, 120.52, 119.18, 66.02 (2C), 45.32 (2C), 41.34, 35.24. HRMS (Q-TOF): calculated for C_22_H_21_N_2_O_6_, 409.1394 [M + H], found 409.1392.

### 4.2. Biological Evaluation

#### 4.2.1. Cell Lines and Cell Culture

Human cervical cancer HeLa, human prostate cancer PC3, human colonic carcinoma HCT116, human lung carcinoma A549, human breast cancer MDA-MB-231, mouse melanoma B16, human ovarian cancer SKOV3, human hepatoma Hep3B, and human liver L02 cells were purchased from the Cell Bank of Shanghai Institute of Biochemistry and Cell Biology, Chinese Academy of Sciences (Shanghai, China). The HeLa, A549, B16, PC3, MDA-MB-231, HCT116, and L02 cells were cultured in Dulbecco’s modified Eagle medium (DMEM, Gibco Inc.; Gaithersburg, MD, USA). All cultures were supplied with 10% fetal bovine serum (FBS; YHSM, Beijing, China), 100 IU/mL of penicillin, and 100 μg/mL of streptomycin, and cells were incubated at 37 °C in an atmosphere of 5% CO_2_.

#### 4.2.2. Cell Counting Kit-8 (CCK-8) Assay

A CCK-8 kit (MCE, USA) was used to perform a CCK-8 assay. Cells were plated and grown in the 96-well plates for the indicated period (48 h) in the incubator, followed by an incubation with the CCK-8 solutions for 1–4 h. Subsequently, the absorbance (450 nm) was measured by a microplate reader. The median inhibitory concentration (IC_50_) was determined from the dose–response curve. Experiments were performed in a triplicate.

#### 4.2.3. Hoechst-33342 Staining

Apoptosis was determined by Hoechst 33342 staining. HeLa cells were seeded in a 6-well plate for 24 h, with 4 × 10^5^ cells per well. Then, the cells were incubated with different concentrations of compound **5c** or 0.1% DMSO for another 24 h. Then, the cells were harvested and stained with Hoechst 33342 for 10 min. The cells were washed with PBS twice and then observed under a fluorescent microscope (LEICA DMI3000 B, Shanghai, China).

#### 4.2.4. Western Blotting Assays 

Protein levels were determined by the standard Western blot assay. HeLa cells were incubated with different concentrations of compound **5c** or 0.1% DMSO for 24 h. Cells harvested with trypsin were treated with 1× RIPA lysis buffer (50 mM Tris-HCl, pH 7.4, 150 mM NaCl, 0.25% deoxycholic acid, 1% NP-40, 1 mM ethylenediaminetetraacetate (EDTA), and protease inhibitor PMSF (Solarbio, Beijing, China) to extract the total proteins. Then, an aliquot of proteins from the total cell lysates (4–40 μg/lane) was separated using sodium dodecyl sulfate (8, 10, or 12%) polyacrylamide gel electrophoresis (BioRad Laboratories, Hercules, CA, USA). The aliquot was wet-transferred to a PVDF membrane (BioRad Laboratories, Hercules, CA, USA), blotted with primary antibodies (purchased from Cell Signaling Technology, Danvers, MA, USA) specific for the STAT3 (#12640), p-STAT3 (#9145), AKT (#4691), p-AKT (#4060), Bax (#5023), Bcl-xl (#2764), secondary isotype-specific antibodies (#7074) and glyceraldehyde-3-phosphate dehydro-genase (GAPDH, #2118) as the internal standard overnight; then, they were probed with secondary isotype-specific antibodies for another 2 h at 37 °C. The bound immuno-complexes were detected using a ChemiDOC XRS+ system (BioRad Laboratories, Hercules, CA, USA). 

#### 4.2.5. Cell Cycle and Annexin V apoptosis Assays

For the cell cycle analysis, HeLa cells seeded in a 6-well plate (3 × 10^5^ per well) were incubated with different concentrations of compound **5c** or 0.1% DMSO for 24 h. Then, the cells were digested with 0.25% trypsin (without EDTA) and were washed twice with PBS, fixed in 70% ethanol overnight, and stained with PI (50 mg/mL, Sigma) plus 0.2 mg/mL of DNase-free RNase A (Qiagen) for 20 minutes at room temperature. For the annexin V apoptosis assay, HeLa cells seeded in a 6-well plate (3 × 10^5^ per well) were incubated with different concentrations of compound **5c** or 0.1% DMSO for 24 h. Then, the cells were digested with 0.25% trypsin (without EDTA) and were washed twice with PBS. The cells were then collected and resuspended in 500 μL of the binding buffer. We added 5 μL of annexin V-FITC and PI each into the mixture in the dark. After 20 min of incubation, a flow cytometer (FACSCalibur; Becton Dickinson, Franklin Lakes, NJ, USA) was used to detect apoptosis (FITC Ex/Em = 488/525 nm and PI Ex/Em = 535/615 nm). All flow cytometric analyses were performed at the Molecular and Cell Platform of Tianfu Science and Technology Park in West China Hospital.

#### 4.2.6. Wound Healing Assay

A total of 1 × 10^6^ HeLa cells/well in the logarithmic growth phase were seeded into 6-well plates. When the cell density reached 80–90%, a scratch was made in the monolayer in the middle of the well with a 200 μL pipette tip. The tip was kept perpendicular to the bottom of the well to obtain a straight gap. The detached cells were washed away and removed. Wound healing within the same scraped line was then observed and photographed at the indicated time points (0 h, 24 h, and 48 h). Images were taken at 0 h, 24 h and 48 h after scraping using a phase-contrast microscope. The wound closure was calculated as a percentage (%) of the difference in the width of the wound between 0 h, 24 h and 48 h relative to 0 h, as described below: Wound closure (%) = (Width of wound at 0 h − Width of wound at 24 h)/Width of wound at 0 h × 100%

#### 4.2.7. Colony Formation Assay

HeLa cells were cultured at 1 × 10^3^ cells per well in a 6-well plate with a regular growth medium for 12 h. After the cells were fixed on the wall, they were treated with different concentrations of compound **5c** or 0.1% DMSO for 24 h. Then, the cells were cultured in a fresh medium for two weeks until the colonies were visible. A crystal violet staining solution (Beyotime, Shanghai, China) was used to stain the colonies for 30 min, and the result was recorded by a smartphone. The colonies were defined as >50 cells/colony.

#### 4.2.8. Cellular ROS Detection In Vitro

To measure reactive oxygen species (ROS) production, 2′,7′-dichlorofluorescein diacetate (DCFDA) (Thermo Fisher Scientific, Waltham, MA, USA; D399) was used [40]. DCFDA is a fluorogenic dye that, after diffusion into the cell, is oxidized by ROS into 2′,7′-dichlorofluorescein (DCF), a highly fluorescent compound that can be detected by fluorescence spectroscopy. To measure ROS production, HeLa cells were treated with compound **5c** at 6 μM, 9 μM, and 12 μM for 6 h. Then, cells were washed with pre-warmed 1X PBS and were incubated at 37 °C with 10 μM of DCFDA for 30 min in PBS. Subsequently, the HeLa cells were washed and analyzed in FL-1 using a FACScalibur flow cytometer (BD, USA) and using CELL Quest software (BD Biosciences, San Jose, CA, USA). Live cells were gated according to their forward scatter (FSC) and side scatter (SSC) properties. For each analysis, 20,000 events were recorded. 

#### 4.2.9. Biolayer Interferometry (BLI) Assay

Purified recombinant STAT3 proteins were biotinylated using the EZ-Link biotinylation reagent (Thermo Fisher Scientific). Briefly, the protein and biotinylation reagent were mixed with a 1:1 molar ratio in PBS at 4 °C. Low biotinylation reagent concentration was applied to avoid protein over-biotinylation. These reaction mixtures were incubated at room temperature for 30 min to allow for the completion of the reaction. Reaction mixture was then dialyzed using 10K MWCO dialysis cassettes (Thermo Fisher Scientific) to remove unreacted biotinylation reagent.

BLI experiments were performed using an OctetRED96 instrument from ForteBio. All assays were run at 25 °C with continuous 1000 RPM shaking. PBS with 0.1% BSA, 0.02% Tween-20, and 2% DMSO was used as the assay buffer. The biotinylated STAT3 proteins were tethered on Super Streptavidin (SSA) biosensors (ForteBio) by dipping the sensors into 1 mg/mL of the protein solutions. Average saturation response levels of 10–14 nm were achieved in 10 min for all STAT3 proteins. Sensors with proteins tethered were washed in assay buffer for 10 min to eliminate nonspecifically bound protein molecules and to establish stable baselines before starting association-dissociation cycles with the test compound. DMSO only references were included in all assays. The raw kinetic data that were collected were processed in the data analysis software provided by the manufacturer using double reference subtraction, in which both the DMSO only and inactive references were subtracted. The resulting data were analyzed based on a 1:1 binding model from which the k_on_ and k_off_ values were obtained. After, the KD values were calculated.

#### 4.2.10. Molecular Docking

A molecular docking study was carried out to dock compound **5c** to predict its binding mode and approximate binding free energy to the STAT3 SH2 domain with the computational docking program Schrodinger. The extended form of the STAT3 structure (PDB: 6NUQ) was used for the docking study. Compound **5c** and the protein were subjected to a standard prepared workflow in the Schrodinger suite. A grid box with the size of 20 × 20 × 20 Å was determined, and the box was centered on a co-crystalized ligand. The center coordinates of grid box were set at x: 3.18, y: 49.35, and z: −1.76; other docking parameters were set as default. After completing 10 million energy evaluations, the root-mean-square deviation threshold was set as 2.0 Å, and all conformations of the ligands in the binding pocket of the macromolecule were clustered. Next, compound **5c** was docked into the SH2 domain using the Glide XP docking procedure, where the lowest energy clusters were identified and the binding energy was evaluated. The results were visualized with Py MOL 2.4.0 software.

#### 4.2.11. Molecular Dynamics Simulation

The complex of compounds **5c** and **SI-109** with STAT3 (PDB code: 6NUQ) was subjected to MDSs in a solution system using the Gromacs 5.1.1 software, parameterized with the Gromacs 96 force field. The protein–ligand complex was placed in a cubic water box. Then, the system was neutralized with sodium chloride counter ions and we optimized the system through the steepest descent algorithm. A position restraint of 1000 kJ·mol^−1^·Å^−2^ was then applied to both the ligand and protein atoms, and then a NVT balance of 500 ps and NPT balance of 500 ps were carried out to restrict the position of the whole system. Finally, the MDS was performed on the well-equilibrated system, and we thus obtained the binding free energy calculation.

### 4.3. Statistical Analysis

Data are presented as the means ± standard deviation (SD). The comparison of quantitative data in multiple groups was performed using a one-way analysis of variance (ANOVA) test followed by the Tukey test using the statistical software GraphPad Prism 6.0 (GraphPad Software, San Diego, CA, USA). *p* < 0.05 was regarded as statistically significant.

## Data Availability

The data presented in this study are available within the manuscript and Appendix A.

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
