# Peer review of "SAR Study and Molecular Mechanism Investigation of Novel Naphthoquinone-furan-2-cyanoacryloyl Hybrids with Antitumor Activity"

_pharmaceutics, 2022, doi:10.3390/pharmaceutics14102104_

Round 1

Reviewer 1 Report

Authors provide the synthesis of novel candidates to antitumor agents, then an exhaustive characterization of their biological activity was performed. (In vitro antitumor activity and safety, inhibition of Cell Migration, inhibition of colony formation, blocking of cells in G2/M phase, generation of reactive oxygen species, cell apoptosis, phosphorylation inhibition, kinetic affinity against STAT3). In addition, a docking experiment was performed.

Manuscript is suitable for publication with major changes, docking results are preliminary and must be improved, in addition, extra computational methods are needed to validate docking results and conclusions.

Minor observations.

1.     Authors must justify the PFC strategy with references, for instance reference: DOI:10.1021/ci800219h, https://doi.org/10.1021/jm701021b, etc.

2.     In figure 3 and las paragraph of introduction section.  The definition of a pharmacophore moiety is commonly given by the superposition of structures with similar biological activity, in such comparison common molecular fragments are the pharmacophore moiety. Please provide references regarding with the pharmacophore model used, or the methodology used to define such pharmacophore model.

3.     Please provide a Supplementary Information file with the confirmation of synthesized molecules (Nuclear Magnetic Resonance, Mass Spectrometry, etc.,).

Major concerns:

1.     Regarding docking, please justify the use of the E-coli Transcription Factor STAT3B to characterize the binding mode of a tested molecule. An E-coli STAT3B structure is used, nevertheless, the biological assays are performed with human cultures.

2.     Please provide a validation for the docking procedure. The PDB: 1BG1 is a STAT3B-DNA complex, authors must dock the reported DNA sequence on STAT3B to obtain a RMSD less than 2 Angstroms to validate the Docking parameters. Please report the size of the grid, the search algorithm, the scoring function, etc.

3.     The reported poses must be accompanied with a quantitative analysis of the binding energies, otherwise, it is possible that a reported pose lacks physicochemical meaning, for instance, a positive binding energy.

4.     Finally, the stability of a ligand-protein complex must be evaluated through a molecular dynamic simulation (at least 100 nano seconds), and a subsequent MM/PBSA analysis to obtain binding and solvation Gibbs free energies. I do strongly recommend to authors to perform such study.

Reviewer 2 Report

The article needs to be examined for these mistakes.

Some points to be considered:

·         Given that this research was conducted inside of a university hospital and that I was unable to locate a lab where the new compounds were prepared, I would like to know specifically in which lab the new compounds were created.

·         Results and discussion part must comprise the elucidation of the chemical structures of the novel synthesized derivatives utilizing various spectrum analyses.

·         A supplemental material file must contain all of the new compounds' spectral data because the supplementary data section could not be identified.

·         For additional confirmation of all the new compounds, IR spectra must be provided.

·         The creation of the novel compounds should be described in order in the experimental section. Since it is illogical to begin with compound number 2, followed by compound number 6, compound number 7, and then the derivatives of compound number 4.

·         What procedure is used to purify the derivatives of compound no. 4? What kind of solvent is being utilized, and is it crystallization or silica gel chromatography?

·         Throughout the entire experimental section, J-coupling needs to be redone in italics. Correcting phrases like "J = 8.2 Hz" to [J = 8.2 Hz] and so on for the other compounds in the experimental portion.

·         In the experimental part, vehicle names for all the compounds must begin with a capital letter. For ex. "5-(1,4-dimethoxynaphthalen-2-yl)furan-2-carbaldehyde" should be changed to "5-(1,4-Dimethoxynaphthalen-2-yl)furan-2-carbaldehyde," and so on..

·         Figure 3: "phamacophores" on the arrow corrected to [pharmacophores].

·         Figure 6: start from the first line

·         "Figure 8A" in line 198, "Figure 8B" in line 202 and "Figure 9" in line 218 put in parentheses.

·         Line 272: " (100mL) corrected to (100 mL).

·         Line 558: “37 °C” corrected to [37°C].

Reviewer 3 Report

Liu et al describe the biological effect of new cinnamic acid benzyl amide quinones on several cancer cell lines. This manuscript is well structure, however, lacks the adequate use of reference in all sections. Additionally, the compounds have not values of efficacy or potency (for 5c) of interest for more detailed studies. In Hela cells, 5c exhibits similar IC50 value that WP1066, lacking the interest on STAT3 inhibition reported. It is not clear the link between increased ROS production and p-STAT3 inhibition by 5c. These effects may be unrelated actions and experiments with antioxidants or ROS scavengers in combination with 5c will reveal a possible mechanism of STAT3 inhibition. These experiments must be included if authors describe to 5c as a STAT3 inhibitor.

I suggest to modify the title of this manuscript, because the compounds are not “potent anticancer agents”, but represent a valuable SAR study on quinones on STAT3 in cancer cells.

In sentence: “Later, a large number of novel cinnamic acid benzyl 63 amide analogs, such as WP1609 and WP1066 (Figure 3), were reported to inhibit cancer cells in vitro and in 64 vivo [15]” (lines 63-64), please indicate the inhibitory effect of these compounds… inhibit the proliferation of cancer cells(?)

Line 65: “protein kinase Rio1”, please indicate the full name of Rio1.

Line 68: “…phase 3 research.” Include the respective reference.

Line 110: “…tively low potency (0.25% - 32.35%).” since IC50 values were not calculed, it is not correct to stat “potency”. Please correct to “low efficacy”.

Line 113: similar observation to Line 110.

In tables 1 and 2: Legend: indicate the time of exposition to the new compounds.

Figure 6: In graph, include the error standard in each condition.

Lines 179-181: These sentences refer to methodology and it is not result. Please remove it.

Lines 185: “Some studies have shown that oxidative stress can inhibit AKT.” Please include the references.

Line 186: “5c effectively inhibited the 185 phosphorylation level of AKT (Figure 7C),” Please indicate the residue that is phosphorylated in Akt.

Line 188: “intracellular reactive oxygen species production.” According the previous paragraph, it must say “intracellular ROS production”.

Figure 7: Include the time of treatment for these experiments.

Line 196: “Apoptosis in cancer cells could be caused by the generation of intracellular ROS.” Include reference.

Line 195: “As many quinone compounds have antiproliferation activity by inhibiting the STAT3,” Include the respective reference about this.

Figure 9: Please indicate the time of treatment for this experimental set.

Line 241: “A large percentage of STAT3 inhibitors reduce STAT3 phosphorylation by occupying the SH2 domain.” Include the references.

Methods:

Section: 4.2.4: Please include the ID of all antibodies used.

All Figures: Include the number of independent experiment used for each figure.

Round 2

Reviewer 1 Report

Authors address all the observations. The manuscript is ready for publication.

Reviewer 2 Report

·    For clerical errors, the manuscript still has to be reviewed.